# *Posyandu* Application for Monitoring Children Under-Five: A 3-Year Data Quality Map in Indonesia

**Fedri Ruluwedrata Rinawan** [1,2,3,*], **Afina Faza** [4,5], **Ari Indra Susanti** [1,2], **Wanda Gusdya Purnama** [6], **Noormarina Indraswari** [1,2], **Didah** [1,2], **Dani Ferdian** [1,2], **Siti Nur Fatimah** [1], **Ayi Purbasari** [6], **Arief Zulianto** [7], **Atriany Nilam Sari** [8], **Intan Nurma Yulita** [9,10], **Muhammad Fiqri Abdi Rabbi** [3,11] and **Riki Ridwana** [3,11]

1   Department of Public Health, Faculty of Medicine, Universitas Padjadjaran, Jalan Ir. Soekarno KM. 21, Jatinangor, Sumedang 45363, Indonesia; ari.indra@unpad.ac.id (A.I.S.); noormarina@unpad.ac.id (N.I.); didah@unpad.ac.id (D.); dani.ferdian@unpad.ac.id (D.F.); siti@unpad.ac.id (S.N.F.)
2   Center for Health System Study and Health Workforce Education Innovation, Faculty of Medicine, Universitas Padjadjaran, Jl. Eyckman No. 38, Bandung 40161, Indonesia
3   Indonesian Society for Remote Sensing Branch West Java, Gedung Labtek IX-C lt.3 Jalan Ganesha 10, Institut Teknologi Bandung, Bandung 40132, Indonesia; muhfiqriabdirabbi@upi.edu (M.F.A.R.); rikiridwana@upi.edu (R.R.)
4   Master of Public Health Study Program, Faculty of Medicine, Universitas Padjadjaran, Jalan Eyckman No. 38 Gedung RSP Unpad Lantai 4, Bandung 40161, Indonesia; afina20005@mail.unpad.ac.id
5   Biomedical Engineering Study Program, School of Electrical Engineering, Telkom University, Jl. Telekomunikasi No. 1, Terusan Buahbatu—Bojongsoang, Sukapura, Dayeuhkolot, Bandung 40257, Indonesia
6   Informatics Engineering Study Program, Faculty of Engineering, Universitas Pasundan, Jl. Dr. Setiabudi No. 193, Bandung 40153, Indonesia; wanda.gusdya@unpas.ac.id (W.G.P.); pbasari@unpas.ac.id (A.P.)
7   Postgraduate Master Study Program of Informatics Engineering, Universitas Langlangbuana, Jl. Karapitan No. 116, Cikawao, Kec. Lengkong, Bandung 40261, Indonesia; madzul@unla.ac.id
8   Midwifery Study Program, Faculty of Medicine, Universitas Sebelas Maret, Jl. Ir. Sutami No. 36, Kentingan, Jebres, Surakarta 57126, Indonesia; atriany.ns@staff.uns.ac.id
9   Department of Computer Science, Faculty of Mathematics and Natural Sciences, Universitas Padjadjaran, Jalan Ir. Soekarno KM. 21, Jatinangor, Sumedang 45363, Indonesia; intan.nurma@unpad.ac.id
10   Research Center for Artificial Intelligence and Big Data, Universitas Padjadjaran, Jalan Ir. Soekarno KM. 21, Jatinangor, Sumedang 45363, Indonesia
11   Geographic Information Science Study Program, Faculty of Social Sciences Education, Universitas Pendidikan Indonesia, Jalan Dr. Setiabudi No. 299, Bandung 40154, Indonesia
*   Correspondence: f.rinawan@unpad.ac.id

**Abstract:** *Posyandu* is an Indonesian mother-child health, community-based healthcare. The provision of the *Posyandu* data quality map is crucial for analyzing results but is limited. This research aimed to (a) demonstrate data quality analysis on its completeness, accuracy, and consistency and (b) map the data quality in Indonesia for evaluation and improvement. An observational study was conducted using the *Posyandu* application. We observed data in Indonesia from 2019 to 2021. Data completeness was identified using children's visits/year. Data accuracy was analyzed using WHO anthropometry z-score and implausible z-score values analyzing the outliers. Cronbach's α of variables was used to know data consistency. STATA 15.1 SE and QGIS 3.10 was used to analyze and map the quality. Data completeness and accuracy in three years show a good start for the pilot project area, continued with declines in pandemic time, while some other areas demonstrated a small start, then slightly increased. The overall consistency decreased through the study period. A good report on data completeness can occur initially in a pilot project area, followed by others. Data accuracy and consistency can decrease during the pandemic. The app can be promising when synchronized with the government health information system.

**Keywords:** data quality; iPosyandu; map; mHealth

## 1. Introduction

Health informatics tools (HIT) using mobile health (mHealth) apps have been widely used because they can collect data from end-users and store them in a source database [1].

Using a healthcare setting results in electronic medical record documentation and database storage [2]. Such documentation is also needed in the community healthcare setting. For example, community health workers (CHWs) use mobile apps rather than laptops because they are easier to use. However, using it is another training issue because technology literacy about documentation and reporting needs much effort [3]. Moreover, access to adequate training in technology literacy during the pandemic is problematic [2,4]. It can cause poor data quality [4–6]. Data quality is a prerequisite for data analysis in medical research, preventing errors, and providing information for evidence-based intervention in health promotion [7,8]. Providing evidence-based policy depends on analysis that must use processed data quality from raw to ready-to-analyzed data. In general, public health data quality in an information system consists of many dimensions. Examples include completeness, accuracy, timeliness or up-datedness, validity, relevance, reliability, comparability, internal or external consistency, and data management [6]. The World Health Organization (WHO) reviewed four dimensions: (1) completeness and timeliness, (2) internal consistency, (3) external consistency, and (4) external comparison [9]. The accuracy analysis observed data with the WHO Child Growth standard formula to get the proportion of plausible values [10].

In Indonesia, a mother-child health (MCH) community healthcare setting has been established since the 1980s, named *Posyandu*, abbreviated from *Pos Pelayanan Terpadu* or Integrated service post [3,11,12]. *Posyandu* is located in every village, run by CHWs, and supervised by a village midwife in coordination with *Pusat Kesehatan Masyarakat* (*Puskesmas*) or the Public Health Center [3,13]. Regarding mHealth, midwives and CHWs have access to MCH's data. Midwives can perform MCH screening, assessment, implementation, and evaluation of midwifery care based on the data [14]. *Posyandu* monthly activities start from registering mothers and their under-five children, measuring weight and height, giving nutritional education, and immunizations. In addition, vitamin A is given twice a year (in February and August) [12]. The CHWs record those activities in their notebook and then rewrite those data into the *Posyandu* information system (PIS) book and the mother-child health (MCH) book. Then, it is given to *Puskesmas*, the first layer of prehospital healthcare facilities for treating health problems. Afterward, data is received by healthcare workers (HCWs), such as midwives, nutrition staff, and nurses in Puskesmas, and reported using the national government application (app), such as a nutrition recording and reporting, called the ePPGBM app. However, documenting and reporting activities from *Posyandu* at the community to the national level is a long and time-consumed process.

To shorten the process, we built an independent mobile health application for *Posyandu*, the iPosyandu app, in 2017. Interoperability with the government's health information system, such as ePPGBM, is the primary goal of building the iPosyandu app. The ePPGBM provides Excel forms that can be imported to it. iPosyandu can export similar Excel forms already filled in with the MCH data needed, then import the forms to the ePPBGM. The iPosyandu app was initiated, implemented, and evaluated in our pilot project area, Purwakarta Regency, West Java. Then the use widened to almost all provinces in Indonesia. However, the data quality needs improvement. Efforts to build capacity for CHWs and mothers of children under five using the app have been conducted since 2018. CHWs need more time to adapt to the app, from registering their identity (id) to the data collection [3] and being supervised by village midwives to obtain good data quality. Thus, the data can be used to improve the analysis results.

Previous research provided healthcare data with geographical information system (GIS) [15–17], and other research depicted environmental health analysis also using GIS [18]. According to International Organization for standardization, mapping geographical data quality comprises categories such as completeness, logical consistency, positional and thematical accuracy, and temporal quality [19]. Providing a community data quality map is crucial before analyzing results using GIS [20], but recent literature reviews stated that it is challenging [21,22], more importantly for our country, Indonesia. A data quality map in nutrition is most commonly derived from regional or national anthropometric data but

lacks household data [23]. Our research provides more detail than household data. We analyzed data quality from individual data, i.e., relating to each child. The data quality map is critical and beneficial for further analysis to build a higher quality information system for social purposes, such as promoting health [24]. Furthermore, data quality is vital for machine learning, GIS, and social remote sensing (remote sensing complemented by social studies) to complete the health information system [25]. The problem that needs to be solved to gain good data quality relates to the person who performs the data entry [26]. A dedicated person for such a task is essential to an organization. However, data entry officers are often in limited numbers in healthcare settings because they focus on health services rather than data entry. In this context, we hypothesize that the data quality has improved over the last three years as CHWs learned how to use the app. Three indicators were analyzed to assess variations in data quality: the data completeness, the proportion of plausible values, and the consistency of collected variables. We also mapped these indicators over the years to see if they evolved similarly across Indonesia.

## 2. Materials and Methods

An observational study was conducted using the *Posyandu* application, named iPosyandu, registered in Google Play in December 2018, as a result of the project starting in Purwakarta regency, West Java/Jawa Barat province. Since then, its use has spread all over Indonesia until now. We observed the data quality for the last three years (2019–2021). We chose 3 of 4 dimensions from the WHO data quality review (DQR) framework: (1) completeness, (2) accuracy, and (3) internal consistency [9]. We did not use the fourth dimension, external consistency, because the iPosyandu back-end has not yet been connected with the government HIT. Thus, we have not had access to it. WHO stated that completeness refers to the percentage of monthly reports in a year. In the context of this research, we define accuracy as the proportion of plausible values in the data (i.e., not outliers, as defined by WHO). Consistency corroborates internal consistency between related data items at different times [3].

Before analysis, data duplications were tested using the identification number (id) of children under five and the date of the visit to *Posyandu*. We want to ensure no more than one data entry per child per month, reflecting one report to *Posyandu* monthly. The reason is that *Posyandu* activity per village is held monthly. Ideally, it is expected to be 12 visits in a year. WHO categorizes reporting data of 9–12 visits per year as the expected complete report of *Posyandu* [9]. Our data completeness was identified using one variable, which has the category of 1–4, 5–8, and 9–12 visits reported per year. The cell percentage of data completeness and accuracy of all provinces were presented in the table results. WHO stated that the timeliness of district reporting should be at least 75% of the monthly reports submitted on time, and received at the national level from the district. The WHO also suggests that the benchmark depends on each country. Our Ministry of Health mentioned 75% as 2022's target [27].

In the context of our research, data accuracy was analyzed using WHO Child Growth Standards to get the proportion of outliers (implausible values) and accurate (plausible values). It has been standardized and widely used, and we use WHO anthropometry standards in the app's back-end. The standard can be installed from the STATA package installer to run the data accuracy using four variables: age, weight, height, and gender. In *Posyandu* activities, these data are collected: age and gender are from registration, then weight and height are recorded in the mother-child health (MCH) book after measurement. The presence of outliers was analyzed by the WHO criteria of implausible z-score value: *waz* (weight per age) > 5 standard deviations (SD) and $<-6$ SD, *haz* (height per age) > 6 and $<-6$, *whz* (weight per height) > 5 and $<-5$ SD [10]. This implausible value is an alarm sign for reassessment whether it is an existing biological confounding or technical reason, such as a human error in data entry [10,28]. The remaining observations are considered plausible values. We did not delete the data duplication to see the data entry behaviors in analyzing the accuracy and consistency. Thus, it may result in a higher total number of data. The Chi-square test was used to analyze the difference in completeness and accuracy in

3 years. We hypothesize that there were differences in completeness and accuracy between years, and this information can support data quality measurement [6].

Data consistency was analyzed using Cronbach Alpha of 5 variables used in iPosyandu: ownership of mother-child health (MCH) book, anthropometry as the result of weight and height measurement, supplementary feeding as the service in *Posyandu* for baby and child, including those who suffer diarrhea, immunizations as the data that need to be recorded in MCH book, and vitamin A that routinely given twice a year. The WHO mentioned four metrics of internal consistency; in this research, we used the coherence between the related data items and at different time points because we do not have access to review of source documents in health facilities [9]. STATA version 15.1 Special Edition License (StataCorp LLC, College Station, Texas 77845, USA) was used for the analysis. QGIS 3.10 (open source) and shapefile of the 34 provinces in Indonesia were used to map the data quality. The community activities in *Posyandu* are participatory empowerment. Community participation can be mapped as social mapping. Social mapping represents a way to visualize the aspects of society in specific periods, such as participatory social patterns over time and the relationship between society and spatial factors. In social mapping, more classes are required to display visual information in each region. We used more classes in the completeness and accuracy maps. If a smaller number of classes is used, the information of each region will not be visible [29,30].

## 3. Results

Table 1 depicts total data completeness with a significant difference from 2019 to 2021, starting from 28.73%, where 11.23% completed their visits (9–12/year) to *Posyandu*. In 2020 (pandemic year 1), the total completeness was 19.31%, of which 11.74% still visited 1–4 times a year. In 2021, it increased to 51.96% (pandemic year 2), with most of the 1–4 and 5–8 visits/year. We checked duplication on the completeness, and we found 3583 (5.36%) duplications, resulting in 63,272 non-duplications (Table 1) from the total data of 66,855 (this total number is stated in Table 2).

**Table 1.** Total data completeness analysis each year.

| Data Completeness * | 2019 | | 2020 | | 2021 | | Total | | *p* |
|---|---|---|---|---|---|---|---|---|---|
| | **n** | **%** | **n** | **%** | **n** | **%** | **n** | **%** | |
| 1–4 | 5951 | 9.41 | 7430 | 11.74 | 15,740 | 24.88 | 29,121 | 46.03 | |
| 5–8 | 5120 | 8.09 | 2124 | 3.36 | 11,192 | 17.69 | 18,436 | 29.14 | 0.000 ** |
| 9–12 | 7106 | 11.23 | 2666 | 4.21 | 5943 | 9.39 | 15,715 | 24.84 | |
| Total | 18,177 | 28.73 | 12,220 | 19.31 | 32,875 | 51.96 | 63,272 | 100.00 | |

\* (visits reported/year). ** Chi-square test.

**Table 2.** Total data accuracy analysis each year.

| Data Accuracy | 2019 | | 2020 | | 2021 | | Total | | *p* |
|---|---|---|---|---|---|---|---|---|---|
| | **n** | **%** | **n** | **%** | **n** | **%** | **n** | **%** | |
| Outliers | 5168 | 7.73 | 4332 | 6.48 | 1988 | 2.97 | 11,488 | 17.18 | |
| Accurate | 13,664 | 20.43 | 8797 | 13.15 | 32,936 | 49.24 | 55,397 | 82.82 | 0.000 * |
| Total | 18,832 | 28.16 | 13,129 | 19.63 | 34,924 | 52.51 | 66,855 | 100.00 | |

\* Chi-square test.

Total data accuracy decreased at the beginning of the pandemic year (2020) and increased in the next year, in which the outliers continuously decreased in these three years. The difference between years is significant (<0.05). This information can be seen in Table 2.

The detail of data quality in every province in the three years is explained below. It is illustrated in Table 3 (the completeness), Table 4 (The accuracy and consistency), and Figures 1–3.

**Table 3.** Data completeness per province 2019–2021.

| Province | Data Completeness 2019 | | | | | Data Completeness 2020 | | | | | Data Completeness 2021 | | | | |
|---|---|---|---|---|---|---|---|---|---|---|---|---|---|---|---|
| | 1–4 | 5–8 | 9–12 | Total | % | 1–4 | 5–8 | 9–12 | Total | % | 1–4 | 5–8 | 9–12 | Total | % |
| Aceh | | | | | | 60 | 0 | 0 | 60 | 0.49 | 4 | 0 | 0 | 4 | 0.01 |
| Bali | 118 | 167 | 112 | 397 | 2.18 | 158 | 30 | 0 | 188 | 1.54 | 131 | 142 | 0 | 273 | 0.83 |
| Banten | 5 | 0 | 0 | 5 | 0.03 | 77 | 0 | 0 | 77 | 0.63 | 477 | 652 | 66 | 1195 | 3.63 |
| Bengkulu | 3 | 0 | 0 | 3 | 0.02 | 46 | 0 | 0 | 46 | 0.38 | | | | | |
| DI Yogyakarta | 104 | 0 | 0 | 104 | 0.57 | 25 | 0 | 0 | 25 | 0.20 | 0 | 5 | 0 | 5 | 0.02 |
| DKI Jakarta | 693 | 0 | 0 | 693 | 3.81 | 1044 | 174 | 1285 | 2503 | 20.48 | 3644 | 2739 | 442 | 6825 | 20.76 |
| Gorontalo | | | | | | 60 | 0 | 0 | 60 | 0.49 | | | | | |
| Jambi | 41 | 89 | 0 | 130 | 0.72 | 26 | 0 | 0 | 26 | 0.21 | | | | | |
| Jawa Barat | 4617 | 4595 | 6721 | 15,933 | 87.65 | 3852 | 1112 | 965 | 5929 | 48.52 | 8961 | 5381 | 4803 | 19,145 | 58.24 |
| Jawa Tengah | 7 | 5 | 0 | 12 | 0.07 | 334 | 0 | 0 | 334 | 2.73 | 1 | 0 | 0 | 1 | 0.00 |
| Jawa Timur | 7 | 0 | 0 | 7 | 0.04 | 99 | 0 | 0 | 99 | 0.81 | 1344 | 824 | 0 | 2168 | 6.59 |
| Kalimantan Barat | | | | | | 34 | 0 | 0 | 34 | 0.28 | 50 | 33 | 0 | 83 | 0.25 |
| Kalimantan Selatan | 3 | 0 | 0 | 3 | 0.02 | 22 | 0 | 0 | 22 | 0.18 | 78 | 0 | 0 | 78 | 0.24 |
| Kalimantan Tengah | 8 | 0 | 0 | 8 | 0.04 | 277 | 88 | 10 | 375 | 3.07 | 93 | 0 | 0 | 93 | 0.28 |
| Kalimantan Timur | 21 | 16 | 57 | 94 | 0.52 | 55 | 43 | 0 | 98 | 0.80 | 72 | 0 | 0 | 72 | 0.22 |
| Kalimantan Utara | | | | | | 33 | 0 | 0 | 33 | 0.27 | 7 | 0 | 0 | 7 | 0.02 |
| Kepulauan Bangka Belitung | 18 | 0 | 0 | 18 | 0.10 | 4 | 0 | 0 | 4 | 0.03 | | | | | |
| Kepulauan Riau | | | | | | 3 | 0 | 0 | 3 | 0.02 | 47 | 186 | 9 | 242 | 0.74 |
| Lampung | 195 | 14 | 0 | 209 | 1.15 | 275 | 314 | 66 | 655 | 5.36 | 258 | 50 | 0 | 308 | 0.94 |
| Maluku | | | | | | 83 | 173 | 0 | 256 | 2.09 | | | | | |
| Maluku Utara | | | | | | | | | | | 44 | 0 | 0 | 44 | 0.13 |
| Nusa Tenggara Barat | 28 | 0 | 0 | 28 | 0.15 | 233 | 80 | 318 | 631 | 5.16 | 135 | 440 | 0 | 575 | 1.75 |
| Nusa Tenggara Timur | | | | | | 7 | 0 | 0 | 7 | 0.06 | | | | | |
| Papua | 26 | 0 | 0 | 26 | 0.14 | | | | | | | | | | |
| Papua Barat | 1 | 0 | 0 | 1 | 0.01 | | | | | | | | | | |
| Riau | 1 | 0 | 0 | 1 | 0.01 | 66 | 0 | 0 | 66 | 0.54 | | | | | |
| Sulawesi Barat | 1 | 0 | 0 | 1 | 0.01 | | | | | | 18 | 128 | 0 | 146 | 0.44 |
| Sulawesi Selatan | | | | | | 139 | 0 | 0 | 139 | 1.14 | | | | | |
| Sulawesi Tengah | 48 | 234 | 216 | 498 | 2.74 | 343 | 110 | 22 | 475 | 3.89 | 167 | 491 | 605 | 1263 | 3.84 |
| Sulawesi Tenggara | 1 | 0 | 0 | 1 | 0.01 | 50 | 0 | 0 | 50 | 0.41 | 16 | 0 | 0 | 16 | 0.05 |
| Sulawesi Utara | | | | | | 3 | 0 | 0 | 3 | 0.02 | 2 | 0 | 0 | 2 | 0.01 |
| Sumatera Barat | 4 | 0 | 0 | 4 | 0.02 | | | | | | | | | | |
| Sumatera Selatan | 1 | 0 | 0 | 1 | 0.01 | 20 | 0 | 0 | 20 | 0.16 | 191 | 121 | 18 | 330 | 1.00 |
| Sumatera Utara | | | | | | 2 | 0 | 0 | 2 | 0.02 | | | | | |
| Total | 5951 | 5120 | 7106 | 18,177 | 100.00 | 7430 | 2124 | 2666 | 12,220 | 100.00 | 15,740 | 11.92 | 5943 | 32,875 | 100.00 |

Red: Highlight for no data available.

**Table 4.** Data accuracy and consistency per province 2019–2021.

| Province | Data Accuracy 2019 | | | | | | c. alpha * | Data Accuracy 2020 | | | | | | c. alpha * | Data Accuracy 2021 | | | | | | c. alpha * |
|---|---|---|---|---|---|---|---|---|---|---|---|---|---|---|---|---|---|---|---|---|---|
| | O ** | % | A ** | % | Total | % | | O ** | % | A ** | % | Total | % | | O ** | % | A ** | % | Total | % | |
| Aceh | | | | | | | no obs | 3 | 0.02 | 57 | 0.43 | 60 | 0.46 | 0.5326 | 1 | 0.00 | 3 | 0.01 | 4 | 0.01 | 0.7273 |
| Bali | 94 | 0.50 | 306 | 1.62 | 400 | 2.12 | 0.0598 | 18 | 0.14 | 174 | 1.33 | 192 | 1.46 | 0.205 | 29 | 0.08 | 250 | 0.72 | 279 | 0.80 | 0.503 |
| Banten | 1 | 0.01 | 4 | 0.02 | 5 | 0.03 | 0.8824 | 32 | 0.24 | 51 | 0.39 | 83 | 0.63 | 0.4205 | 173 | 0.50 | 1217 | 3.48 | 1390 | 3.98 | 0.1952 |
| Bengkulu | 3 | 0.02 | 0 | 0.00 | 3 | 0.02 | too few | 54 | 0.41 | 0 | 0.00 | 54 | 0.41 | too few | | | | | | | no obs |
| DI Yogyakarta | 51 | 0.27 | 61 | 0.32 | 112 | 0.59 | 0.7353 | 3 | 0.02 | 23 | 0.18 | 26 | 0.20 | 0.8679 | 0 | 0.00 | 5 | 0.01 | 5 | 0.01 | too few |
| DKI Jakarta | 199 | 1.06 | 558 | 2.96 | 757 | 4.02 | 0.7356 | 275 | 2.09 | 2416 | 18.40 | 2691 | 20.50 | 0.6014 | 299 | 0.86 | 7060 | 20.22 | 7359 | 21.07 | 0.6565 |
| Gorontalo | | | | | | | no obs | 63 | 0.48 | 0 | 0.00 | 63 | 0.48 | 0.7441 | | | | | | | no obs |
| Jambi | 130 | 0.69 | 0 | 0.00 | 130 | 0.69 | no obs | 26 | 0.20 | 0 | 0.00 | 26 | 0.20 | too few | | | | | | | no obs |
| Jawa Barat | 4306 | 22.87 | 12,155 | 64.54 | 16,461 | 87.41 | 0.5462 | 2615 | 19.92 | 3828 | 29.16 | 6443 | 49.07 | 0.5101 | 1044 | 2.99 | 19,222 | 55.04 | 20,266 | 58.03 | 0.4122 |
| Jawa Tengah | 12 | 0.06 | 3 | 0.02 | 15 | 0.08 | 0.6074 | 18 | 0.14 | 334 | 2.54 | 352 | 2.68 | 0.7326 | 0 | 0.00 | 2 | 0.01 | 2 | 0.01 | 1 |
| Jawa Timur | 6 | 0.03 | 1 | 0.01 | 7 | 0.04 | 0.748 | 36 | 0.27 | 67 | 0.51 | 103 | 0.78 | .*** | 117 | 0.34 | 2179 | 6.24 | 2296 | 6.57 | .*** |
| Kalimantan Barat | | | | | | | no obs | 19 | 0.14 | 24 | 0.18 | 43 | 0.33 | 0.4244 | 15 | 0.04 | 72 | 0.21 | 87 | 0.25 | 0.2757 |
| Kalimantan Selatan | 0 | 0.00 | 3 | 0.02 | 3 | 0.02 | not valid | 5 | 0.04 | 18 | 0.14 | 23 | 0.18 | 0.6652 | 3 | 0.01 | 88 | 0.25 | 91 | 0.26 | 0.8374 |
| Kalimantan Tengah | 5 | 0.03 | 3 | 0.02 | 8 | 0.04 | 0.8019 | 128 | 0.97 | 264 | 2.01 | 392 | 2.99 | 0.3987 | 6 | 0.02 | 88 | 0.25 | 94 | 0.27 | 0.0673 |
| Kalimantan Timur | 33 | 0.18 | 87 | 0.46 | 120 | 0.64 | 0.5636 | 6 | 0.05 | 120 | 0.91 | 126 | 0.96 | 0.519 | 2 | 0.01 | 74 | 0.21 | 76 | 0.22 | 0.6064 |
| Kalimantan Utara | | | | | | | no obs | 4 | 0.03 | 29 | 0.22 | 33 | 0.25 | 0.4386 | 1 | 0.00 | 6 | 0.02 | 7 | 0.02 | 0.525 |
| Kepulauan Bangka Belitung | 10 | 0.05 | 9 | 0.05 | 19 | 0.10 | 0.7008 | 2 | 0.02 | 3 | 0.02 | 5 | 0.04 | 0.9701 | | | | | | | no obs |
| Kepulauan Riau | | | | | | | no obs | 2 | 0.02 | 1 | 0.01 | 3 | 0.02 | 0.9231 | 1 | 0.00 | 241 | 0.69 | 242 | 0.69 | 0.3328 |
| Lampung | 140 | 0.74 | 79 | 0.42 | 219 | 1.16 | 0.201 | 103 | 0.78 | 574 | 4.37 | 677 | 5.16 | 0.299 | 82 | 0.23 | 231 | 0.66 | 313 | 0.90 | 0.0934 |
| Maluku | | | | | | | no obs | 81 | 0.62 | 227 | 1.73 | 308 | 2.35 | 0.493 | | | | | | | no obs |
| Maluku Utara | | | | | | | no obs | | | | | | | no obs | 11 | 0.03 | 35 | 0.10 | 46 | 0.13 | 0.6502 |
| Nusa Tenggara Barat | 19 | 0.10 | 9 | 0.05 | 28 | 0.15 | 0.322 | 519 | 3.95 | 138 | 1.05 | 657 | 5.00 | 0.6449 | 115 | 0.33 | 467 | 1.34 | 582 | 1.67 | 0.3561 |
| Nusa Tenggara Timur | | | | | | | no obs | 2 | 0.02 | 5 | 0.04 | 7 | 0.05 | 0.7653 | | | | | | | no obs |
| Papua | 1 | 0.01 | 27 | 0.14 | 28 | 0.15 | too few | | | | | | | no obs | | | | | | | no obs |
| Papua Barat | 1 | 0.01 | 1 | 0.01 | 2 | 0.01 | too few | | | | | | | no obs | | | | | | | no obs |
| Riau | 0 | 0.00 | 1 | 0.01 | 1 | 0.01 | not valid | 68 | 0.52 | 0 | 0.00 | 68 | 0.52 | 1 | | | | | | | no obs |
| Sulawesi Barat | 0 | 0.00 | 1 | 0.01 | 1 | 0.01 | not valid | | | | | | | no obs | 10 | 0.03 | 150 | 0.43 | 160 | 0.46 | 0.137 |
| Sulawesi Selatan | | | | | | | no obs | 39 | 0.30 | 100 | 0.76 | 139 | 1.06 | 0.6474 | | | | | | | no obs |
| Sulawesi Tengah | 155 | 0.82 | 351 | 1.86 | 506 | 2.69 | 0.7022 | 189 | 1.44 | 290 | 2.21 | 479 | 3.65 | 0.2494 | 51 | 0.15 | 1218 | 3.49 | 1269 | 3.63 | 0.1985 |
| Sulawesi Tenggara | 2 | 0.01 | 0 | 0.00 | 2 | 0.01 | not valid | 18 | 0.14 | 33 | 0.25 | 51 | 0.39 | 0.4947 | 2 | 0.01 | 14 | 0.04 | 16 | 0.05 | 0.5813 |
| Sulawesi Utara | | | | | | | no obs | 0 | 0.00 | 3 | 0.02 | 3 | 0.02 | too few | 0 | 0.00 | 2 | 0.01 | 2 | 0.01 | 1 |
| Sumatera Barat | 0 | 0.00 | 4 | 0.02 | 4 | 0.02 | too few | | | | | | | no obs | | | | | | | no obs |
| Sumatera Selatan | 0 | 0.00 | 1 | 0.01 | 1 | 0.01 | not valid | 3 | 0.02 | 17 | 0.13 | 20 | 0.15 | 0.486 | 26 | 0.07 | 312 | 0.89 | 338 | 0.97 | 0.204 |
| Sumatera Utara | | | | | | | no obs | 1 | 0.01 | 1 | 0.01 | 2 | 0.02 | 1 | | | | | | | no obs |
| Total | 5168 | 27.44 | 13,664 | 72.56 | 18,832 | 100.00 | 0.56 | 4332 | 33 | 8797 | 67.00 | 13,129 | 100.00 | 0.49 | 1988 | 5.69 | 32,936 | 94.31 | 34,924 | 100.00 | 0.44 |

\* c.alpha = Cronbach alpha coefficient; no obs= no observation; \*\* O = Outliers, A = Accurate Data \*\*\* = no result ouput in STATA, only (.). Red: Highlite for no data available; Yellow: Highlight for non-zero % Outliers; Green: Highlight for non-zero % Accurate data.

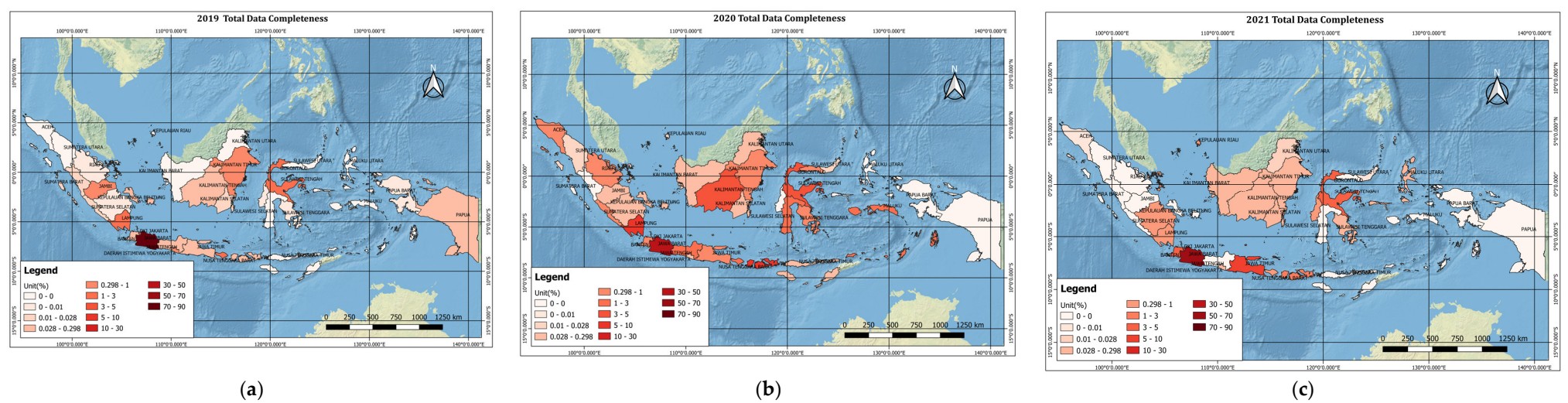

**Figure 1.** Total data completeness figure from 2019–2021: (**a**) 2019; (**b**) 2020; (**c**) 2021.

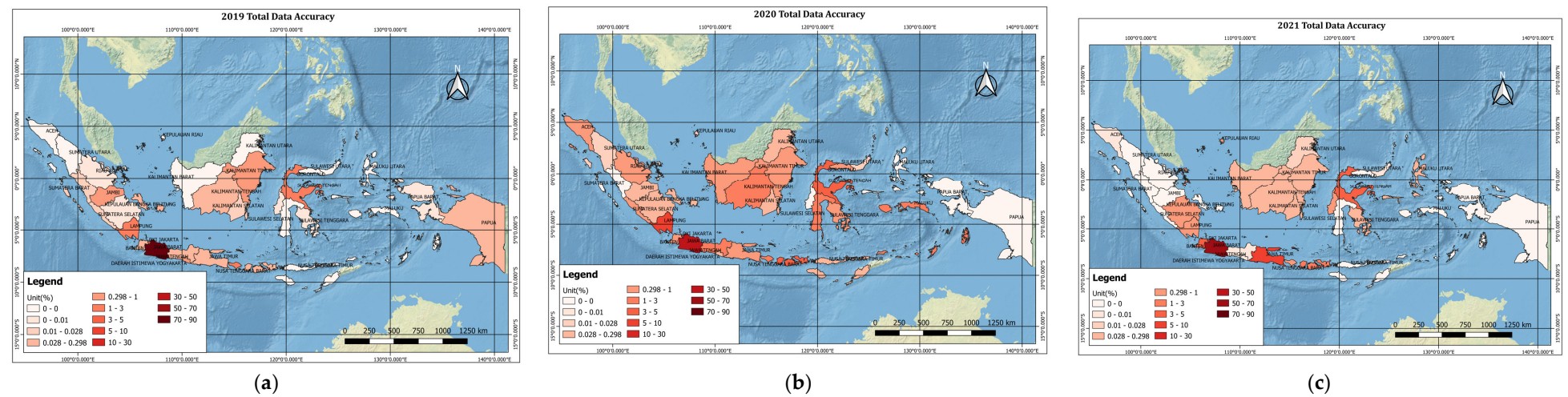

**Figure 2.** Total data accuracy figure from 2019–2021: (**a**) 2019; (**b**) 2020; (**c**) 2021.

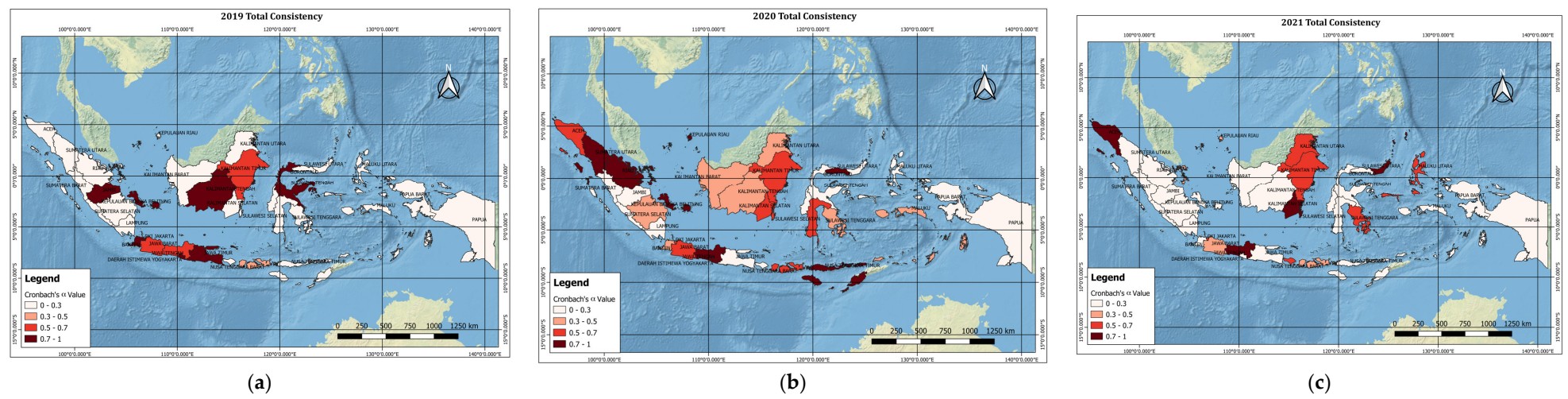

**Figure 3.** Data consistency figure from 2019–2021: (**a**) 2019; (**b**) 2020; (**c**) 2021.

### 3.1. Data Completeness

Table 3 shows data completeness per province from 2019 to 2021. We highlight some emerging results from the total data completeness. We found good reporting data on completeness (87.68%) in West Java (Jawa Barat) but low in Jakarta (3.8%), Central Sulawesi/Sulawesi Tengah (2.73%), Bali (2.18%), Lampung (1.15%), and the rest of the provinces. Generally, complete data in almost all provinces decreased within three years. West Java is the province with the most profound decreased gap between 2019 and the following pandemic years, whereas Central Sulawesi, which started low, slightly increased in the following years. The red highlight on the table refers to no data available because the CHWs in the province did not use the app. We used the other highlights to make the reader easier to locate the completeness of use. The maps in Figure 1a–c demonstrate the total completeness comprising the summary of the number of visits to *Posyandu*: 1–4, 5–8, and 9–12 visits/year. The spread of the app use completeness decreases in spatial distribution (Figure 1) but is higher in percentage than in the previous years (51.96%, $p < 0.05$ in Table 1). The community healthcare activities, including data entry, decreased during the pandemic, and some areas in the country concentrated on improving the number of *Posyandu* reports on mother and child visits, including after the pandemic.

### 3.2. Data Accuracy and Consistency

Table 4 illustrates data accuracy and consistency per province from 2019- to 2021. We found that West Java holds the highest accuracy percentage because this is the pilot project province. Other provinces that participated in using the app, such as Jakarta, Central Sulawesi, East Java, and Bali, were lower in accuracy percentages. We used the red, yellow, and green highlight colors with a similar function as in the data completeness section. Figure 2a–c shows the total data accuracy within the three years, generally similar to the completeness. The accuracy of areas like Jakarta, Central Sulawesi, and East Java is increasing, albeit the pandemic. Although the spatially higher spread area (white part, not accurate) through the years (Figure 2), data accuracy inclined on the existing colored provinces in 2021 (Table 4).

The overall consistency in each year was 0.56, 0.49, and 0.45, respectively. On the map (Figure 3a–c), we used classifications of high consistency (>0.7–1), medium (>0.5–0.7), low (0,3–0,5), not consistent (<0.3) [31,32]. We detailed the 'not consistent' listed on the map's legend, so we do not lose the information. In Table 4, we can see that higher consistency can occur on a small number of data and the other way around. The not available data part (red on the data accuracy column or "no obs" on the consistency column) can be interpreted as people not using the app.

## 4. Discussion

Our data quality (completeness, accuracy, and consistency), which is still in the first three years, demonstrates good quality for the pilot project area but low in others. Data quality of health informatics tools may still be challenging for long-term use. Some applications need eight years to reach good quality. It is promising that consistent app development with training, e.g., using tutorial videos on the app, is vital [33]. Choosing one area is vital for a pilot project in development [34]. After registering in Google Play, when focusing on user-friendliness implemented in the regency, an app can endow easier usability for end-users in other areas in the same country. The main factors embedded in the end-users for data quality are digital literation, motivation, and characteristics (education level, age, profession) [33,35,36].

The criteria of the cadre can shift to younger productive ages because this chooses for longer working life in community healthcare [37]. Previous research stated that the younger aged person could coach the older aged person, creating higher capabilities for the older aged person to perform in digital health [38]. However, older cadres have more mature working experiences than younger ones [39]. In our pilot project area, we found that age was not related to their knowledge in running the mHealth because the learning

process can occur at either younger or older [40]. The younger the age, the higher the digital literacy and the other way around. Nevertheless, regardless of age, training and motivation can increase the service and data entry [3,40,41]. The support factor from the midwife also plays an essential role in monitoring, supervising, and verifying the inputted data by the cadre using mHealth. This collaborative support can motivate the cadre to use mHealth [42]. Thus, inputting and reporting data from CHWs to midwives through mHealth can be faster and on time [43]. Both roles (of midwife and cadre) are interrelated for the satisfaction of the cadre in using the app. Collaboration of them is a cornerstone in running the consistency of *Posyandu* [3,11]. Determinant factors that affect the use of applications include perceived benefits, obstacles, and the appearance of easy applications [44]. User comfort is the main thing in achieving satisfaction. There are five dimensions in measuring application satisfaction: content dimensions [45], data accuracy dimensions [46,47], display dimensions [48], ease of use dimensions [49], and timeliness [50].

On the community level, the presence of a village midwife can coach and supervise the cadre when performing service, data documentation, and report. In the first level of the healthcare center in Indonesia, called, *Puskesmas*, data can be retrieved, checked, and verified by the healthcare workers (HCWs) before sending it to the regency health office and Ministry of Health. Regarding the transition from manual to digital, on the one hand, this transition can slow the cadre's process of using the app for service, data recording, and reporting. However, the length of the process is around 2–3 years, enough for development and knowledge diffusion before going faster [3]. The local government can "push" the performance of healthcare workers and their motivation to support CHWs [37] by educating mothers, performing screening, e.g., weight and height, and record them, and referring suspected individuals to village midwives [3]. Governments support their work by giving material rewards, such as incentives [37]. More importantly, providing an internet quota is vital in supporting the gadget [51]. In addition, decentralization from the national government to the local government can make more flexible budget allocations depending on the local condition [52].

The Indonesian government has initiated a human development cadre (HDC) program in 2020/2021 focusing on social mapping through *Posyandu*, such as on nutritional (malnutrition, stunting, wasting) and environmental (lavatory availability) problems, service, and data recording, even though fewer cadres have been recruited [53]. From these nutritional and environmental factors in *Posyandu*, social mapping can be generated as a map with an administrative borderline to illustrate the information from each area, e.g., village, district, city, regency, and province boundaries [53,54]. In this case, the social mapping can be supported by a data quality map in our research. Through the iPosyandu application, cadres can directly input the data resulting from the *Posyandu* activities, whose data can be directly downloaded and verified on data quality. The government e-PPBGM (electronic Community-Based Nutrition Recording and Reporting) app provides front-end integration that the similar Excel file result can be uploaded into the app [55]. So far, the implementation of data input in the e-PPBGM application is still carried out by *Puskesmas* officers based on measurements made by the cadres at the *Posyandu*. Therefore, through the use of iPosyandu by the cadres, this problem can overcome the problem of delays in data input in the e-PPBGM application [3].

Performing data quality maps can be a basis for further analysis, such as machine learning, remote sensing, and geographical information systems (RS-GIS). The machine learning method uses data to build an intelligent system. Data quality is the main course of this method for creating a model [56]. However, raw data is not clean [57]. Not-clean data comprises inconsistency, incompleteness, duplication, inaccuracy, and irrelevance. Experts can preprocess and validate such data before implementing machine learning [58]. In the sense of RS-GIS, data quality maps can support further correlation analysis to environmental factors. Implementing remote sensing technology can determine the environmental factors such as land cover and vegetation associated with human aspects such as diseases.

Geographical information systems can positively impact health-service quality and help stakeholders place an excellent policy [15]. Community-health data quality in a country is better created on a map to be more understandable spatially about the data distribution on which province has higher data quality and which one needs improvement. More importantly, if it is in the form of WebGIS that it can be more accessible [20].

## 5. Conclusions

Good quality data is essential for data analysis in providing information for evidence-based health policy for public health. We analyzed the data quality from iPosyandu for the last three years (2019–2021) using 3 dimensions: completeness, accuracy, and internal consistency. It shows a good report on data completeness initially in a pilot project area, followed widely in other areas in Indonesia. The pilot project area is essential for building and refining health informatics tools. Other areas can voluntarily follow it because the refining process makes it more user-friendly. However, several factors influence the data quality, such as a pandemic can decrease the *Posyandu* activities causing hindrances in data entry and reporting. It also severely affects data accuracy and consistency. The app can be promising when synchronized with the government health information system. The limitation of our study is that the identification (id) number of the cadre who input the data is not yet connected with their data input activities. The id number is vital for interpreting whether the data is duplicated by the same or different cadre, by which further improvement can be added in the app tutorial. Another limitation was that the cutoff of implausible values was not yet added to the app, which influenced our data quality. Further work on the app should improve these limitations.

## 6. Patents

The iPosyandu application copyright has been registered since 2018 with the number 000103655 in Indonesia's Ministry of Law and Human Rights.

**Author Contributions:** Writing, methodology, draft preparation, Fedri Ruluwedrata Rinawan, Afina Faza; conceptualization, Fedri Ruluwedrata Rinawan, Ari Indra Susanti, Dani Ferdian; software; database, data preparation, Wanda Gusdya Purnama, Ayi Purbasari, Intan Nurma Yulita; investigation of app security and functionality, Arief Zulianto, Wanda Gusdya Purnama; data validation, analysis, Siti Nur Fatimah, Noormarina Indraswari; data curation, formal analysis, Wanda Gusdya Purnama, Fedri Ruluwedrata Rinawan, Afina Faza, Noormarina Indraswari; writing—original draft preparation, Fedri Ruluwedrata Rinawan, Afina Faza, Ari Indra Susanti, Noormarina Indraswari, Intan Nurma Yulita, Riki Ridwana, Didah, Dani Ferdian, Atriany Nilam Sari; visualization, Fedri Ruluwedrata Rinawan, Afina Faza, Muhammad Fiqri Abdi Rabbi, Riki Ridwana; data mapping, Muhammad Fiqri Abdi Rabbi, Riki Ridwana; project advocacy, Dani Ferdian, Ari Indra Susanti, Fedri Ruluwedrata Rinawan; project administration, funding acquisition, Ari Indra Susanti, Didah, Fedri Ruluwedrata Rinawan; capacity building of end-users, Ari Indra Susanti, Wanda Gusdya Purnama, Didah, Atriany Nilam Sari, Noormarina Indraswari, Fedri Ruluwedrata Rinawan; writing—review, editing, and supervision, Fedri Ruluwedrata Rinawan. All authors have read and agreed to the published version of the manuscript.

**Funding:** This research was funded by the Lecturer Competence Internal Grant of Universitas Padjadjaran, Indonesia, grant number 2476 /UN6.C/LT/2018; PT. Astra International Tbk, grant number 087/LOA-ESR/VII/2019; the Indonesia Endowment Fund for Education abbreviated LPDP (*Lembaga Pengelola Dana Pendidikan*); the Ministry of Finance, grant number PRJ-70/LPDP/2019; and the Kreasi Insani Persada Foundation, grant number (-).

**Institutional Review Board Statement:** The study was conducted in accordance with the Declaration of Helsinki and approved continuously by the Research Ethics Committee Universitas Padjadjaran (protocol code 830/UN6.C10/PN/2017, 1456/UN6.KEP/EC/2019, 433/UN6.KEP/EC/2020, 640/UN6.KEP/EC/2021, and 160/UN6.KEP/EC/2022).

**Informed Consent Statement:** We used secondary data from the iPosyandu app database. The iPosyandu users must read the terms and conditions, including the informed consent, and then agree to involve in the study before using the app. The database is under the copyright protection of Law in the Republic of Indonesia.

**Data Availability Statement:** Not applicable.

**Acknowledgments:** The authors acknowledge the Purwakarta Regency Office for permission to conduct the app's research and development, including integrating the iPosyandu app into the government health information system. We also acknowledge all *Posyandu*, community healthcare center/*Puskesmas*, and regency/city health offices in Indonesia that have used the app to support their work for supporting community health in Indonesia; and Izma Maulana Ahmad Lugina from the Geographic Information Science Study Program, Faculty of Social Sciences Education, Universitas Pendidikan Indonesia for contributing his help to the map creation.

**Conflicts of Interest:** The authors declare no conflict of interest, and the funders had no role in the design of the study, in the collection, analyses, or interpretation of data, in the writing of the manuscript, or in the decision to publish the results.

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
