# Peer review of "Posyandu Application for Monitoring Children Under-Five: A 3-Year Data Quality Map in Indonesia"

_ijgi, doi:10.3390/ijgi11070399_

Round 1

Reviewer 1 Report

The objective of the research is to assess the data quality obtained from a new application (iPosyandu). The app registers health data from mothers and children living in Indonesian rural areas.

Overall Comments

The expected content of the manuscript would be of interest for any developer that aims at implementing a health-related application that is to be used in a community environment.

However, the manuscript lacks adequate context and sufficient information. The research objectives are clearly stated but remain incomprehensible to the reader due to this lack of context. The reader must make many assumptions about the author objectives, and the nature of the data used in the analyses. Consequently, it was not possible to judge the relevance of the statistical analyzes carried out, as well as to assess the validity of the results. Furthermore, the discussion section is problematic because many of the arguments are not supported by the data, or analysis results presented in previous sections.

Detailed comments are provided in an annotated version of the manuscript uploaded with this report. Minor comments are in yellow, major in red.

Author Response

Reviewer 1:

Comments and Suggestions for Authors

The objective of the research is to assess the data quality obtained from a new application (iPosyandu). The app registers health data from mothers and children living in Indonesian rural areas.

Overall Comments

The expected content of the manuscript would be of interest for any developer that aims at implementing a health-related application that is to be used in a community environment.

Response:

Thank you for the feedback. We are improving it to make it a better app.

However, the manuscript lacks adequate context and sufficient information. The research objectives are clearly stated but remain incomprehensible to the reader due to this lack of context. The reader must make many assumptions about the author objectives, and the nature of the data used in the analyses. Consequently, it was not possible to judge the relevance of the statistical analyzes carried out, as well as to assess the validity of the results. Furthermore, the discussion section is problematic because many of the arguments are not supported by the data, or analysis results presented in previous sections.

Response:

Thank you for the inputs. We just edited as requested here and also in the PDF file.

Detailed comments are provided in an annotated version of the manuscript uploaded with this report. Minor comments are in yellow, major in red.

Response:

Thank you for the very detailed comments. We gave response also to the PDF comments, and edited as requested in the PDF file.

Reviewer 2 Report

This study analyzed the quality of data collected from the Posyandu app, which was developed by the authors. From the reviewer’s point of view, this manuscript is written as an analysis report than a scientific paper. Despite of the reported analysis results, this paper has the following major issues:

1.       In the Abstract, the authors just plainly listed the method, results and conclusion using different sections. Especially for the results, the authors just pasted all the numbers instead of explaining the main findings. Please rewrite the abstract in a scientific manner.

2.       There is a lack of literature on data quality (completeness, accuracy, consistency), and analysis methods used for the evaluation of data quality, especially for geospatial data.

3.       Please explain your data, e.g., how many variables in your data? what are they? Please also give one example sample to illustrate your test data.

4.       The Methods are not introduced properly. For example, the authors state “Data completeness was identified using the category of 9-12 visits/year as complete (C), 5-8 less complete (LC), and 1-4 as not complete (NC).” However, it is unclear why you use “visits/year” and why you choose these numbers for the classification. Are there any reasons behind? Please explicitly explain this. Similarly, please explain (why) the methods used for data accuracy and consistency.

Author Response

Reviewer 2

Comments and Suggestions for Authors

This study analyzed the quality of data collected from the Posyandu app, which was developed by the authors. From the reviewer’s point of view, this manuscript is written as an analysis report than a scientific paper. Despite of the reported analysis results, this paper has the following major issues:

Response:

Thank you for the feedback. We tried to improve and make it sounds more scientific than before.

  1. In the Abstract, the authors just plainly listed the method, results and conclusion using different sections. Especially for the results, the authors just pasted all the numbers instead of explaining the main findings. Please rewrite the abstract in a scientific manner.

Response:

We just removed the numbers and replaced them with some explanation to get more insight from the main findings.

  1. There is a lack of literature on data quality (completeness, accuracy, consistency), and analysis methods used for the evaluation of data quality, especially for geospatial data.

Response:

We added the geospatial data quality information, including social mapping data in the manuscript.

  1. Please explain your data, e.g., how many variables in your data? what are they? Please also give one example sample to illustrate your test data.

Response:

We added them as requested in the Materials and Methods section.

  1. The Methods are not introduced properly. For example, the authors state “Data completeness was identified using the category of 9-12 visits/year as complete (C), 5-8 less complete (LC), and 1-4 as not complete (NC).” However, it is unclear why you use “visits/year” and why you choose these numbers for the classification. Are there any reasons behind? Please explicitly explain this. Similarly, please explain (why) the methods used for data accuracy and consistency.

Response:

Thank you for the detailed correction. We added reasons to support the method.

Reviewer 3 Report

The paper addresses an interesting topic but the background to the approach, theoretical support and empirical methods are largely flawed or left incomplete by the authors. There is no detail given on the use of Posyandu application and its merits over the existing ones and what matrix/model/parameter is used by this approach should have been provided before its application. In addition, if this application is used, what added value is generated in contrast to similar apps/methods. Many of the statements are out of blue and don't enjoy literature support. Similarly, the most of the results also lack reference to similar studies or why they are different from the existing ones. The conclusion section is confusing and need to focus work's limitations, future perspective on research and policy etc. 

Author Response

Reviewer 3:

Comments and Suggestions for Authors

The paper addresses an interesting topic but the background to the approach, theoretical support and empirical methods are largely flawed or left incomplete by the authors. There is no detail given on the use of Posyandu application and its merits over the existing ones and what matrix/model/parameter is used by this approach should have been provided before its application. In addition, if this application is used, what added value is generated in contrast to similar apps/methods. Many of the statements are out of blue and don't enjoy literature support. Similarly, the most of the results also lack reference to similar studies or why they are different from the existing ones. The conclusion section is confusing and need to focus work's limitations, future perspective on research and policy etc.

Response:

Thank you for the compliment and also feedback to make this manuscript understandable. We added information over the manuscript as requested in this feedback.

Round 2

Reviewer 1 Report

The objective of the research is to assess the data quality obtained from a new application (iPosyandu). The app registers health data from mothers and children living in Indonesian rural areas.

Overall Comments

The authors improved their manuscript as suggested, which made it easier to identify some shortcomings in the paper.

The main problem lies in the use of the terms “accurate data” and “data accuracy”. In the text, the term “accurate data” seems to describe data that have not been dismissed as having implausible z-score values [1], which is very confusing. Similarly, the term “data accuracy” seems to refer to the number of observations that are considered as “accurate data”, using the author’s terminology. While the percentage of implausible z-score values is an important indication of data quality [1], the remaining “plausible” observations are simply “observations”, not “accurate observation” nor “accurate data”. Consequently, the whole manuscript (text, tables and maps) must be reviewed accordingly.

Another issue is that many statements used WHO documents as reference [8, 24] but most of them were found to be inaccurate after examination. I am then very suspicious of how the other cited references have been used.

Finally, a less important problem, is the structure of the Introduction and Materials and Methods sections. The description of the data is succinctly provided in the introduction, while the reader would benefit from it being appropriately detailed in the Materials and Methods section. This section would also benefit from being split according to the different analyzes used by the authors, with a clear description of each objective. I provided the authors with a link [2] to a scientific paper writing guide that I still use.

The work done (iPosyandu app) deserves to be known by ensuring that many readers are made aware of app development and evolution. This can only be done through a rigorous and easy-to-read paper. My comments aim to motivate the authors to write the manuscript for these readers.

Detailed comments are provided in an annotated version of the manuscript uploaded with this report. Minor comments are in yellow, major in red, highlights in green.

[1] World Health Organization, UNICEF (2019), Recommendations for collecting and analyzing data and reporting information on anthropometric indicators in children under 5 years of age. Geneva. Section 3.1.5. Authors’ references: [24]

[2] https://www.nature.com/scitable/topicpage/scientific-papers-13815490/

Author Response

Thank you for the feedback, we just improved the manuscript. We also answered the comment in pdf, in the attachment.

Reviewer 2 Report

This manuscript has been significantly improved. Nevertheless, there are still three main issues to be addressed in further revisions.

First, in the Introduction, more relevant and state-of-the-art work should be added. For example, besides the WHO definition of data quality, what are the more general definitions of data completeness, accuracy, and completeness and their applications in the health care domain;  and are there any existing quality mapping methods and results relevant to this study?

Second, in Materials and Methods, everything is mixed into one long paragraph. This should be restructured into several smaller paragraphs, e.g., according to the logic or workflow, from data and data processing, analysis methods, mapping methods, and tools.

Third, the conclusion is written mostly in a kind of technical and analytical way. A more general summary of the (analysis and mapping) methods and workflow and their usefulness at the beginning of the paragraph should be added.

Some minor issues:

1.       For the abstract, I would suggest writing in just one normal paragraph by removing the words, like “Methods”, “Results” and “Conclusions”. Furthermore, the first sentence “A good report on data completeness can occur initially in a pilot project area, followed by others. Data accuracy and consistency can decrease during the pan-43 demic.” in “Conclusions” seems similar to the first sentence in “Results”, which should be removed or rephrased.

2.       On Page 2 Line 77, the sentencerewrite those data Posyandu information system (PIS) book.” should be “rewrite those data into …”

3.       At the beginning of “Materials and Methods”, on Page 3 Line 124, the years when the data was collected and used in this study should be added.

Author Response

Thank you for the feedback, we just improved our manuscript and answer the detail in the attachment.

Reviewer 3 Report

Authors have extensively addressed the comments and made the article much clearer and lucid.

Author Response

  • Thank you for the feedback. We are glad to hear that.

Round 3

Reviewer 1 Report

The objective of the research is to assess the data quality obtained from a new application (iPosyandu). The app registers health data from mothers and children living in Indonesian rural areas.

Overall Comments

The authors have improved their manuscript as suggested with one exception. The problem still lies in the use of the terms “accurate data” or “data accuracy” which refer to the number of observations that have not been excluded as outliers (“implausible” values according to WHO terminology). The remaining “plausible” observations are simply “observations”, not “accurate observation” nor “accurate data”. Consequently, the whole manuscript (text, tables and maps) should have been reviewed accordingly, which was not done.

The structure of the text could still be improved. I provided the authors with a link [1] to a scientific paper writing guide that I still use. As I wrote previously, the work done (iPosyandu app) deserves to be known through a good and easy-to-read paper. My comments aim to motivate the authors to structure the paper for these readers.

Detailed comments are provided in an annotated version of the manuscript uploaded with this report. Minor comments are in yellow, major in red, highlights in green.

[1] https://www.nature.com/scitable/topicpage/scientific-papers-13815490/

Author Response

Thank you for the detailed feedback. We responded to the PDF feedback and we attach it below.
